# EA-UNet Based Segmentation Method for OCT Image of Uterine Cavity

**Zhang Xiao** [1,2,†], **Meng Du** [2,3,†], **Junjie Liu** [1], **Erjie Sun** [1], **Jinke Zhang** [4], **Xiaojing Gong** [4,*] and **Zhiyi Chen** [2,3,5,*]

1 College of Mechanical Engineering, University of South China, Hengyang 421001, China
2 Institute of Medical Imaging, Hengyang Medical School, University of South China, Hengyang 421001, China
3 The First Affiliated Hospital, Medical Imaging Centre, Hengyang Medical School, University of South China, Hengyang 421001, China
4 Research Center for Biomedical Optics and Molecular Imaging, Shenzhen Key Laboratory for Molecular Imaging, Guangdong Provincial Key Laboratory of Biomedical Optical Imaging Technology, CAS Key Laboratory of Health Informatics, Shenzhen Institutes of Advanced Technology, Chinese Academy of Sciences, Shenzhen 518055, China
5 The Seventh Affiliated Hospital, Hunan Veterans Administration Hospital, Hengyang Medical School, University of South China, Changsha 410000, China
\* Correspondence: xj.gong@siat.ac.cn (X.G.); zhiyi_chen@usc.edu.cn (Z.C.)
† These authors contributed equally to this work.

**Abstract:** Optical coherence tomography (OCT) image processing can provide information about the uterine cavity structure, such as endometrial surface roughness, which is important for the diagnosis of uterine cavity lesions. The accurate segmentation of uterine cavity OCT images is a key step of OCT image processing. We proposed an EA-UNet-based image segmentation model that uses a U-Net network structure with a multi-scale attention mechanism to improve the segmentation accuracy of uterine cavity OCT images. The E(ECA-C) module introduces a convolutional layer combined with the ECA attention mechanism instead of max pool, reduces the loss of feature information, enables the model to focus on features in the region to be segmented, and suppresses irrelevant features to enhance the network's feature-extraction capability and learning potential. We also introduce the A (Attention Gates) module to improve the model's segmentation accuracy by using global contextual information. Our experimental results show that the proposed EA-UNet can enhance the model's feature-extraction ability; furthermore, its MIoU, Sensitivity, and Specificity indexes are 0.9379, 0.9457, and 0.9908, respectively, indicating that the model can effectively improve uterine cavity OCT image segmentation and has better segmentation performance.

**Keywords:** optical coherence tomography; image segmentation; deep learning; mechanism of attention

## 1. Introduction

The uterine cavity's anatomical information, including intrauterine area, endometrial thickness, and the endometrial surface's fine structure, has important applications in the diagnosis of obstetric and gynecological diseases and the preoperative evaluation of assisted reproductive technologies. However, suitable imaging means in clinical practice are lacking. The commonly used transvaginal ultrasound reflects coarse information, while hysteroscopy is deficient in reflecting fine structures. Optical coherence tomography (OCT) is a well-developed biomedical imaging technique [1], which has significant advantages, such as real-time, non-invasive, and high-resolution images [2,3]. Combined with OCT intraluminal imaging's application scenario, there is a better prospect of its clinical application for evaluating intraluminal lesions, which already have mature applications in the cardiovascular system [4–7], esophagus [8–10], gastrointestinal tract [11–15], and reproductive tract system [16–21].

OCT images of the uterine cavity can provide clear information regarding its structure, such as endometrial surface roughness, which can be analyzed by OCT image processing.

Recently, a series of studies reported on the application of OCT intracavitary imaging for the female reproductive tract, particularly intrauterine imaging [22]. These reports confirm that OCT can accurately reflect the uterine cavity's structural information at the tissue level of the endometrial surface, which is important for the diagnosis of uterine cavity lesions. However, the accurate segmentation of uterine cavity OCT images is key. The use of computer-aided technology [23–25] (CAD) is an effective method to solve this problem. Early CAD techniques used traditional machine learning segmentation methods based on manual features, and despite some achievements in the field of OCT image segmentation, there are persisting problems, such as a heavy reliance on manually designed features, low feature levels, high computational cost, and a complex processing flow during image processing. In recent years, deep learning technical approaches [26] have achieved remarkable results in numerous computer vision [27–29] fields and medical image analysis applications [30–32]. Moradi et al. [33] proposed an attention-based UNet model for the automatic image analysis, pattern recognition, and segmentation of kidney OCT images. Liu et al. [34] proposed an enhanced nested UNet architecture (MDAN-UNet) by taking advantage of multi-scale input, multi-scale side output, and dual-attention mechanisms. This is a new powerful full-convolutional network for automatic end-to-end OCT image segmentation. Fang et al. [35] proposed a new segmentation framework combining CNN and graph search methods (CNN-GS) to segment the nine retinal layer boundaries in OCT images of patients with non-exudative AMD. It consisted of two main parts: (1) CNN layer boundary classification; (2) CNN probability map-based graph search layer segmentation. Wang et al. [36] used CNN to segment CNV in OCT angiography. Shah et al. [37] proposed a convolutional neural network (CNN)-based framework to segment multiple surfaces simultaneously. A single CNN was trained to segment three retinal surfaces in two OCT images, namely normal retina and retina affected by intermediate age-related macular degeneration (AMD). Chen et al. [38] proposed the multiscale double-attention MSDA-UNet network, an MSDA mechanism network for OCT lesion region segmentation. This network can extract lesion region information from OCT images of different scales and perform an end-to-end segmentation of OCT retinal lesion regions. Santos et al. [39] proposed CorneaNet, a deep fully convolutional neural network for segmenting corneal OCT images with high accuracy. Guo et al. [40] proposed a lightweight network model for segmenting retinal vessels by introducing spatial attention in U-Net, naming it as Spatial Attention U-Net (SA-UNet). Xu et al. [41] proposed an optimized compression excitation connection (SEC) module integrated with UNet called SEC-UNet, which not only focused on the target but also stepped out of the local optimum to obtain accurate and complete results of choroidal layer segmentation in OCT images. Singh et al. [42] presented a new benchmark segmentation of the extraretinal limiting membrane (ELM) using image datasets for spectral domain OCT scans of a population of patients with idiopathic total macular lacunae. Gao et al. [43] proposed a new privileged modal distillation (PMD) framework for VBDI. PMD transforms the single-input single-task (SIST) learning problem in single-mode VBDI into a multi-input multi-task (MIMT) problem to help the learning model in the target modality. Medical image segmentation via deep learning has shown several advantages over traditional machine learning algorithms, including higher accuracy and reliability, more efficient GPU-based computing power, and lower power consumption [44–47]. Convolutional neural networks (CNNs) [48] significantly improve the performance of segmentation tasks by utilizing fast and reliable training. Lee et al. [49] applied CNN to achieve the automatic segmentation of macular edema in OCT images. Fully convolutional neural networks (FCN) [27] have achieved remarkable results in the field of image segmentation. Ronneberger et al., inspired by the FCN network, proposed UNet [50], which combines deep semantic and spatial information through encoder and decoder blocks and jump connections. The UNet architecture has achieved excellent results in many medical image segmentation tasks and is widely used in OCT image segmentation tasks, such as optic nerve papillary tissue segmentation [51], vitreous wart segmentation [52], intraretinal cystic fluid (IRC) segmentation [53], fluid

region segmentation [54], and retinal layer segmentation [55]. Kepp et al. [56] proposed a deep learning algorithm with a deep learning model using a U-Net-based network to segment different tissue regions in OCT images of mouse skin, and the segmentation results were in agreement with the expert manual segmentation results.

At present, there are many clinical applications for various OCT image segmentation methods; however, an effective OCT image segmentation process to gather information from the uterine cavity has not yet been found, and our paper is the first time to propose it. Secondly, the uterine cavity can be segmented accurately using deep learning OCT image processing, which helps to analyze the uterine cavity structure's information, including thickness changes, which is important for the diagnosis of uterine cavity lesions. Finally, our study aims to explore a real-time and accurate uterine cavity OCT image segmentation method. These are the main motives of this study. However, in the feature-extraction process, there is a large amount of redundancy in OCT images, and significant features are easily ignored, resulting in the loss of useful information. Additionally, due to the inconsistent information about the uterine cavity's variations in size and thickness, we must ensure that segmentation remains effective, which requires the model to have high-level feature-processing capabilities. These are the challenges faced in our study.

U-Net is a common architecture applied in medical image segmentation tasks, and it can obtain high-quality results on a limited training set of medical images. In our paper, we combine a multi-scale attention mechanism and propose an encoder–decoder architecture, which we call EA-UNet, for uterine cavity OCT image segmentation. The main contributions of our study are:

First, we adopted the E(ECA-C) module, which introduces the convolutional layer combined with the attention mechanism ECA [57] instead of max pool, which reduces the loss of feature information, enables the model to focus on the features in the region to be segmented, and suppresses the irrelevant features to enhance the model's feature-extraction ability.

Second, we improved the U-Net network structure to optimize the upsampled feature-layer channels and retain more detailed features. Then, inspired by the structure of Attention U-Net [47], we introduced the Attention Gates module to extract features containing more detailed information using global contextual information to enhance the model's detailed segmentation effect and improve its segmentation accuracy.

## 2. Materials and Methods

### 2.1. Experimental Environment and Data

We used Windows 10 Professional, 32 G RAM, NVIDIA RTX 3080 with 10 GB video memory for our experimental environment platform. We adopted the deep learning frameworks PyTorch 1.8.1, cuda11.1, and Python 3.8.5. Our experimental batch size was 2, with 90 rounds of training iterations. Our optimizer used RMSprop, and we set the initial learning rate and weight decay factor to 0.001 and $1 \times 10^{-8}$, respectively.

We carried out all procedures involving experimental animals in accordance with the protocols approved by the animal study committee of the Shenzhen Institute of Advanced Technology, Chinese Academy of Sciences. We purchased four healthy female New Zealand rabbits (4–4.5 kg) from Kangda biological Co., Ltd. (Qingdao, China). We maintained all animals under a 12/12 light/dark cycle at 21–24 °C with a relative humidity of 40–60% and fed with rabbit chow and water ad libitum [22]. According to studies on ethanol-induced apoptosis [58,59], an injection of 95% ethanol into the uterine horn during general anesthesia causes endometrial damage. The extent of endometrial damage depends on the residence time of ethanol in the uterine horn. To establish a rabbit model of endometrial injury, we injected 95% ethanol into the rabbits' left uterine horns for 5 or 10 min, then extracted and slowly rinsed with saline to remove residual ethanol before injecting an equal amount of saline into the contralateral uterine horn. The dataset we used consists of clinical data from endometrial images of rabbits collected by the Shenzhen Institute of Advanced Technology of the Chinese Academy of Sciences using medical OCT equipment [22]. Each image in the dataset has a corresponding label image, and each label image is manually

labeled by two experts with clinical experience as the segmentation standard Ground Truth images, and then reviewed and gated by senior experts in the field of expertise to correct missing or incorrect labeling when appropriate.

### 2.2. Data Augmentation

First, we performed an initial image screening and quality assessment of the dataset images; then, we preprocessed each OCT image in the dataset before the model training and testing experiments. We cropped irrelevant areas and reduced the images to an appropriate size to ensure the main feature information of the OCT images; then, all training set images were disordered and subjected to random flip and other operations before being imported into the training model. After processing, the OCT dataset consisted of a total of 1347 images, including 1007 images of a normal uterine cavity and 340 images of a damaged uterine cavity; then, we divided the dataset into training and test sets according to the ratio of 8:2.

### 2.3. Evaluation Metrics

In order to evaluate the proposed model's segmentation effect, we used *MIoU*, Sensitivity, and Specificity as OCT image segmentation evaluation metrics; the calculation formulas are:

$$MIoU = \frac{1}{k+1} \sum_{i=0}^{k} \frac{TP}{TP + FP + FN} \tag{1}$$

$$Sensitivity = \frac{TP}{TP + FN} \tag{2}$$

$$Specificity = \frac{TN}{TN + FP} \tag{3}$$

where *k* is the total number of categories of segmentation. *TP* denotes true positive, which is the number of pixels inside the predicted region and inside the true-labeled region; *TN* denotes true negative, which is the number of pixels outside the predicted region and the true-labeled region; *FP* denotes false positive, which is the number of pixels inside the predicted region and outside the true-labeled region; *FN* denotes false negative, which is the number of pixels outside the predicted region and inside the true-labeled region.

### 2.4. EA-UNet Network Model

Our proposed multi-scale attention EA-UNet network structure is composed of the proposed ECA-C and Attention Gates modules introduced on the traditional U-Net network, as shown in Figure 1. The EA-UNet's structure, which is an end-to-end segmentation model, includes encoding and decoding regions. First, adding the ECA-C module to the encoding region can make the model focus more on the focus region; second, we introduced the convolutional layer Conv on the attention mechanism ECA to extract more image feature information instead of max pool. Then, we introduced the Attention Gates module between the jump connections to connect the corresponding encoding and decoding areas to prevent the loss of image feature information, improving the model's feature-extraction ability and perfecting the images' feature information. Finally, we changed the number of channels in each layer of the decoding area to recover as many image features as possible and output the segmented image.

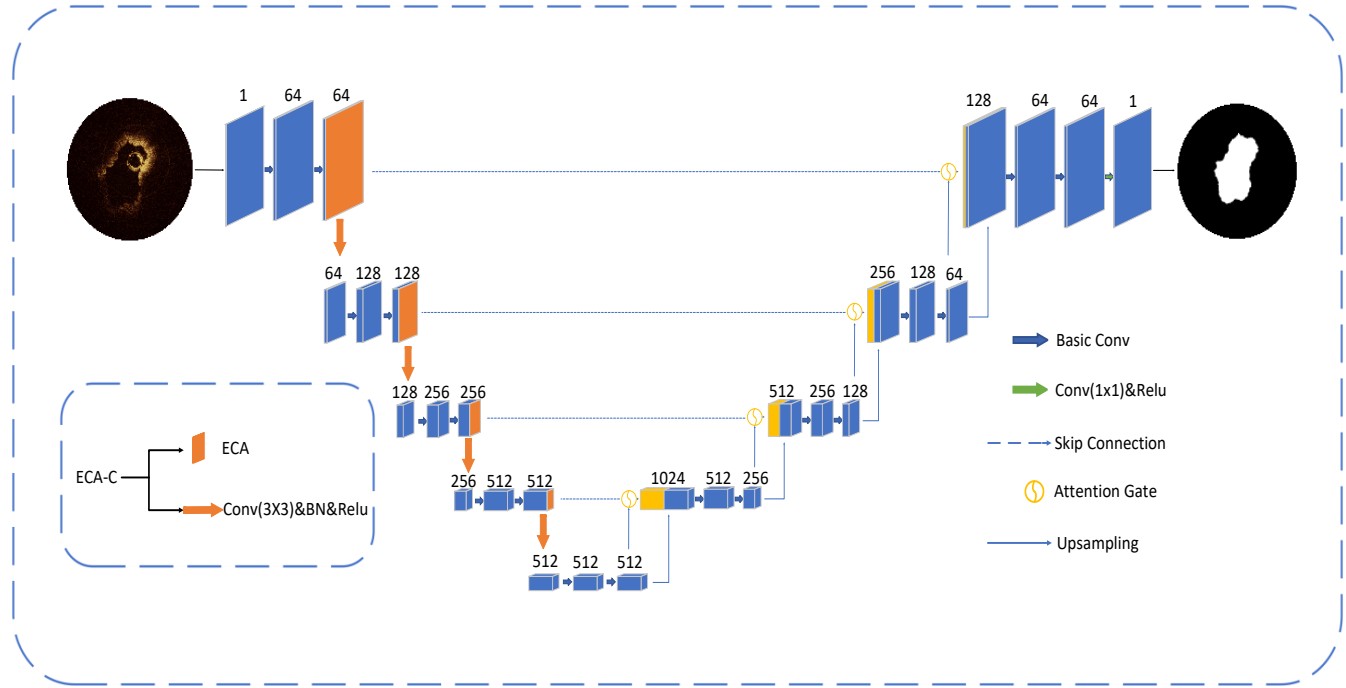

**Figure 1.** EA-UNet network structure.

*2.5. ECA-C Module*

The ECA-C module mainly makes some improvements to the SE-Net [60] module, changing a small number of parameters but achieving considerable performance gains. It is a local cross-channel interaction strategy without dimensionality reduction and a method of adaptively selecting the size of one-dimensional convolution kernels, thereby achieving performance improvements. Avoiding dimensionality reduction is important for learning channel attention, and by properly capturing local cross-channel interactions, one can considerably reduce model complexity while maintaining performance. Therefore, it can be efficiently implemented by one-dimensional convolution with a local cross-channel interaction strategy without dimensionality reduction. According to our experiments, it is efficient and feasible to select the ECA-C module, as shown in Figure 2.

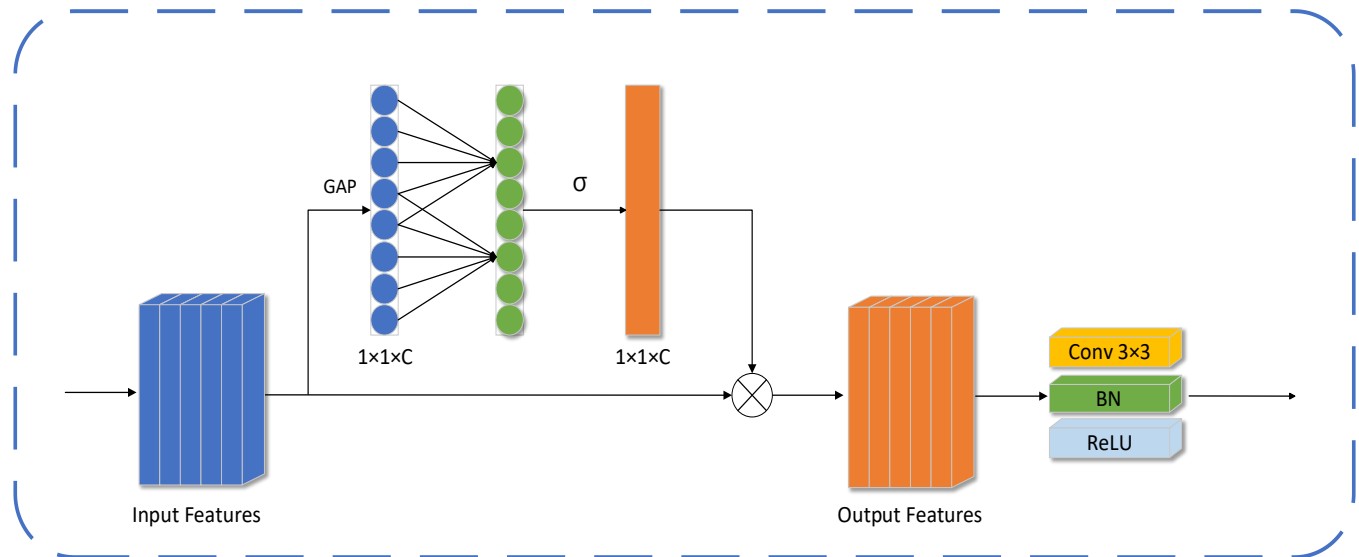

**Figure 2.** ECA-C module structure diagram.

We chose the ECA module to ensure computational performance and model complexity. Among them, $W_k$ is used to denote the learned channel attention, $W_k$ involves K×C parameters, and $W_k$ avoids complete independence of different groups, and for the weight $y_i$, this paper only considers the information interaction between $y_i$ and its $K$ neighbors, i.e.,

$$w_i = \sigma(\sum_{j=1}^{k} w_i^j y_i^j), \ y_i^j \in \Omega_i^k \tag{4}$$

The equation captures local cross-channel interactions, and this locality constraint avoids interactions across all channels, thus allowing for higher model efficiency. To further reduce model complexity and improve efficiency, it is also possible to have all channels share weight information, i.e.,

$$w_i = \sigma(\sum_{j=1}^{k} w^j y_i^j), \ y_i^j \in \Omega_i^k \tag{5}$$

In addition, the ECA module can implement information interactions between channels through a one-dimensional convolution with a convolution kernel size of $K$.

$$w = \sigma(C1D_k(y)) \tag{6}$$

Among them, $C1D$ stands for one-dimensional convolution, which only involves $K$ parameter information. This method of capturing local cross-channel information interactions ensures performance results and model efficiency.

### 2.6. Attention Gates Module

The AG model can automatically respond to feature regions without localizing them and at the same time simulate the position relationship of global feature regions, suppress irrelevant feature responses, enhance similar region features, and then extract image features containing more detailed information to improve the model's segmentation accuracy. Figure 3 shows the input deep feature map $X_g$ and shallow feature map $X_i$ are features summed by one-dimensional convolution to enhance the feature regions, and then the feature image $T_l$ is obtained by the ReLU nonlinear activation function. We performed one-dimensional convolution on $T_l$ to reduce the computational effort and obtained the weight map $\alpha$ (whose element values range from [0, 1]) by resampling after processing by the Sigmoid activation function. It is then multiplied with the feature map $X_i$ to obtain $\widehat{x}^l$ to enhance the image feature representation and attenuate the non-image feature response.

The module uses the semantic information in the deep feature map to enhance the feature weights in the shallow feature map, thus adding more details to the shallow feature map to enhance the model's learning ability for the segmentation region, further improving the model's segmentation accuracy. The calculation method is as follows:

$$T_l = \sigma_1(W_x^T x_i + W_g^T m(g_i) + b_1) \tag{7}$$

$$\alpha_i = \sigma_2(W^T T_l + b_2) \tag{8}$$

$$\widehat{x}^l = x_i \alpha_i \tag{9}$$

among them, $X_i$ and $X_g$ are the shallow and deep feature maps, respectively, $W_x$, $W_g$, and $W^T$ are linear transformation parameters, $b_1$ and $b_2$ are offset terms, and $\sigma_1$ and $\sigma_2$ are the RelU and Sigmoid activation functions, respectively.

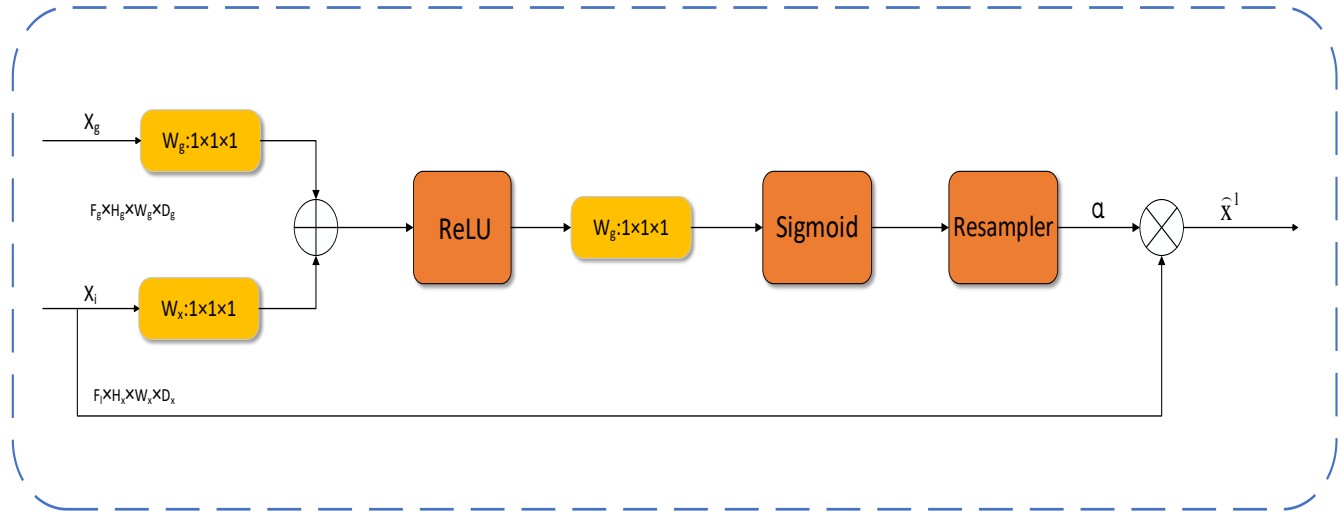

**Figure 3.** Attention Gates module structure diagram.

*2.7. Loss Function*

The binary cross-entropy loss function can be applied to most pixel-level segmentation tasks; however, it can also mislead the model when the number of pixels on the target is far less than the number of pixels in the background [61]. In addition, the Sigmoid layer is integrated into the CELoss class using the BCEWithLogitsLoss loss function. This is numerically more stable than using a simple Sigmoid layer and BCELoss. The formula is as follows:

$$\mathrm{L_{BCE}} = -\sum_{i=1}^{N}[y_i \log p(y_i) + (1 - y_i) \log(1 - p(y_i))] \tag{10}$$

where $y_i$ is the pixel real category label and $p(y_i)$ is the pixel prediction category label.

## 3. Results

### 3.1. Ablation Experiment

The ECA-C module and AG module performances were verified by comparing the segmentation effects of U-Net, Attention + UNet, ECA-C + UNet, and EA-UNet on OCT images (Table 1; bolded indicates the best results).

**Table 1.** Ablation experiments of effect of different modules on performance of U-Net model.

| Model | MIoU | Sensitivity | Specificity |
|---|---|---|---|
| U-Net | 0.8787 | 0.8886 | 0.9865 |
| ECA-C + UNet | 0.9096 | 0.9226 | 0.9831 |
| Attention + UNet | 0.9187 | 0.9343 | 0.9803 |
| EA-UNet | **0.9379** | **0.9457** | **0.9908** |

The experimental results in Table 1 show that U-Net with the addition of the Attention and ECA-C modules improves the original U-Net's segmentation performance for uterine cavity OCT images in two metrics, *MIoU* and Sensitivity; however, the Specificity metric decreases. Our proposed EA-UNet model with both ECA-C and Attention modules has the highest segmentation effect on uterine cavity OCT images.

We visualized the segmentation results of each module in Table 1 to more clearly show the advantages of using the EA module to segment image details. Figure 4 shows the segmentation results of the U-Net model with different Attention modules from Table 1. Four randomly selected OCT images from the test set are presented, where column 1 is the original OCT image, Ground Truth denotes the true label, and columns 2 to 5 denote

the segmentation results of each U-Net, Attention + UNet, ECA-C + UNet, and EA-UNet model, respectively.

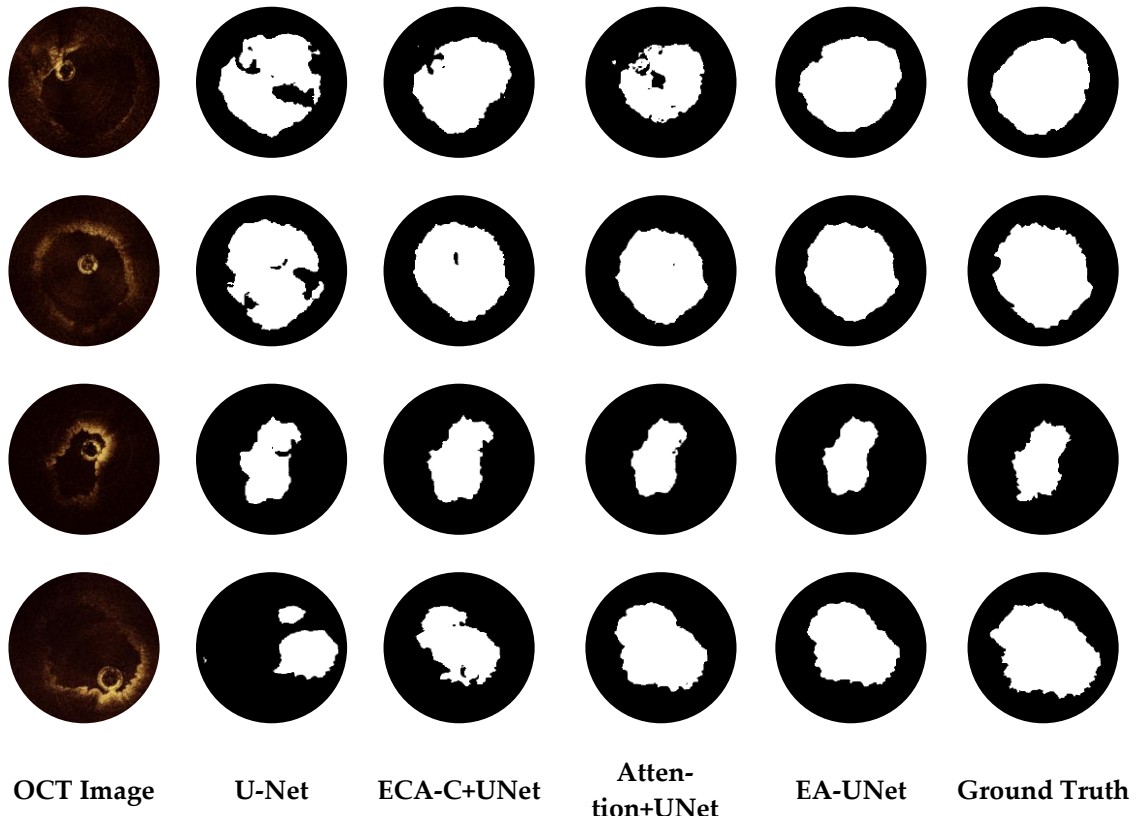

|OCT Image | U-Net | ECA-C+UNet | Atten-tion+UNet | EA-UNet | Ground Truth |

**Figure 4.** Comparison of ablation experiments of each module of EA-UNet model.

The visualization of each model's segmentation results in Figure 4 shows that the U-Net model has the lowest segmentation ability, with obvious problems of blurred boundaries and incomplete image segmentation. The ECA-C + UNet and Attention + UNet models have incomplete segmentation results and rough boundaries, and some detailed regions are ignored. The EA-UNet model, on the other hand, has the highest segmentation effect and can effectively segment the regions, and its segmentation results are closest to the Ground Truth results.

### 3.2. Experiments Comparing EA-UNet Model and Other Attention Methods

To verify the proposed EA-UNet model's effectiveness in uterine cavity OCT image segmentation, it was compared with other attention modules for experiments (Table 2; bolded indicates the best results).

**Table 2.** Performance comparison between EA-UNet and other attention models.

| Model | MIoU | Sensitivity | Specificity |
|---|---|---|---|
| U-Net [50] | 0.8787 | 0.8886 | 0.9865 |
| SCSE + UNet [62] | 0.8391 | 0.8502 | 0.9830 |
| SE + UNet [60] | 0.9051 | 0.9222 | 0.9774 |
| CBAM + UNet [63] | 0.9219 | 0.9399 | 0.9776 |
| EA-UNet | **0.9379** | **0.9457** | **0.9908** |

Table 2 shows the proposed EA-UNet module can achieve optimal results in uterine cavity OCT image segmentation compared with other attention modules. The SCSE attention module not only does not improve the segmentation effect of U-Net but also degrades

its performance. Although the SE and CBAM attention mechanisms improve the MIoU of U-Net metrics, the Specificity metrics are inferior to the original U-Net model. Therefore, the proposed EA-UNet model can achieve effective segmentation in uterine cavity OCT image segmentation.

To demonstrate the proposed EA-UNet model's experimental effects compared with other attention models, we randomly selected four images from the test set to visualize the segmentation results. Column 1 shows the four original OCT images from the test set, Ground Truth indicates the true label, and columns 2 to 6 show the U-Net, SCSE + UNet, SE + UNet, CBAM + UNet, and EA-UNet models' segmentation results, respectively, as shown in Figure 5.

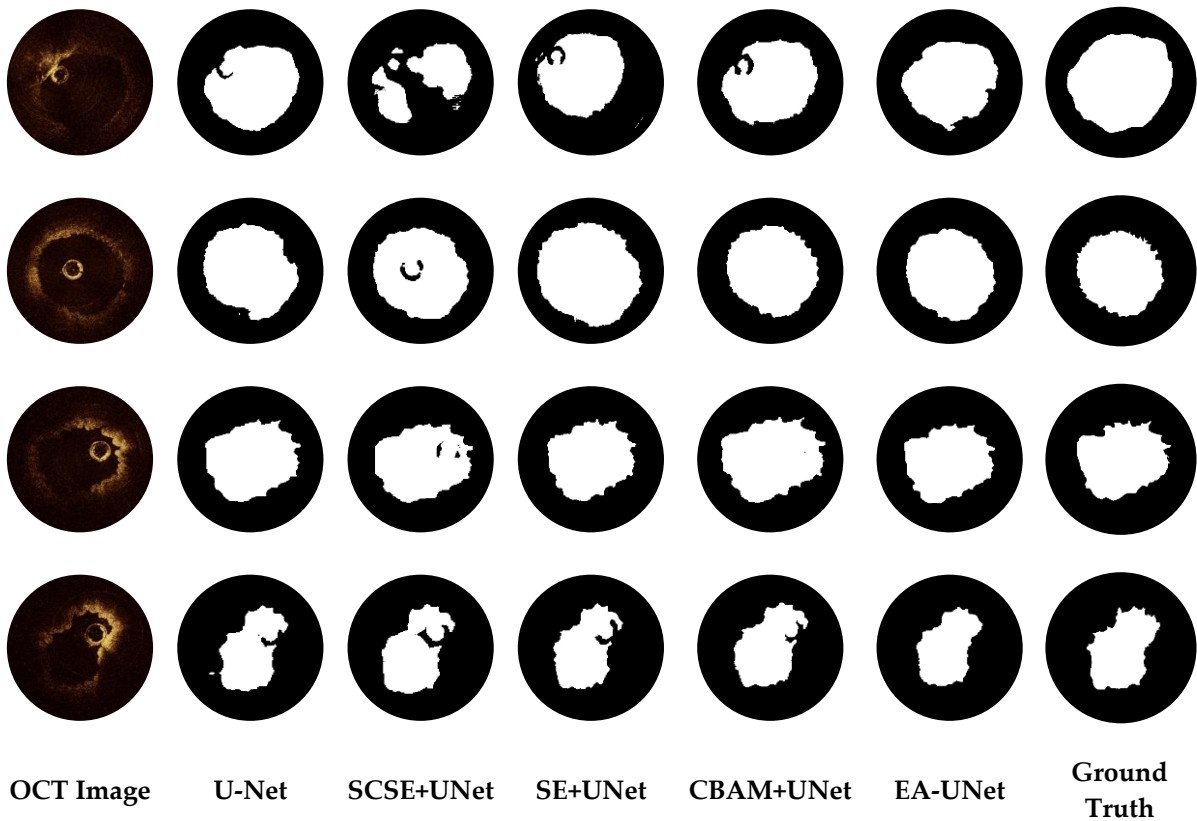

**OCT Image**  **U-Net**  **SCSE+UNet**  **SE+UNet**  **CBAM+UNet**  **EA-UNet**  **Ground Truth**

**Figure 5.** Comparison of segmentation effects of different attention modules.

The visualization of the segmentation results of each model in Figure 5 shows that the SCSE + UNet model has the worst segmentation effect and is blurred. The SE + UNet and U-Net models have incomplete segmentation results, rough boundaries, and some detailed regions are ignored. The CBAM + UNet model's segmentation results are second only to the EA-UNet model. Compared with other attention modules, the EA-UNet model has better feature-extraction ability and learning potential due to the E (ECA-C) module, which introduces a convolutional layer on the attention mechanism ECA instead of max pool, thus reducing the loss of feature information, making the model focus on the features of the region to be segmented and suppressing irrelevant features to enhance the network's feature-extraction ability and learning potential, ensuring a high-quality detail segmentation ability and the best segmentation effect, which can effectively segment the region; furthermore, its segmentation result is closest to the Ground Truth. In addition, we introduced the A (Attention Gates) module to improve the model's full understanding of local contextual information, thereby improving the model's segmentation accuracy. Meanwhile, the loss function fitting curve shown in Figure 6 is derived based on our experimental results. Compared with the other four models, when training under the same experimental conditions, our proposed method decreases the loss value faster with little

fluctuation during training, and the loss rate is smaller, meaning the model has a faster convergence speed during training, which indicates that the method has good robustness.

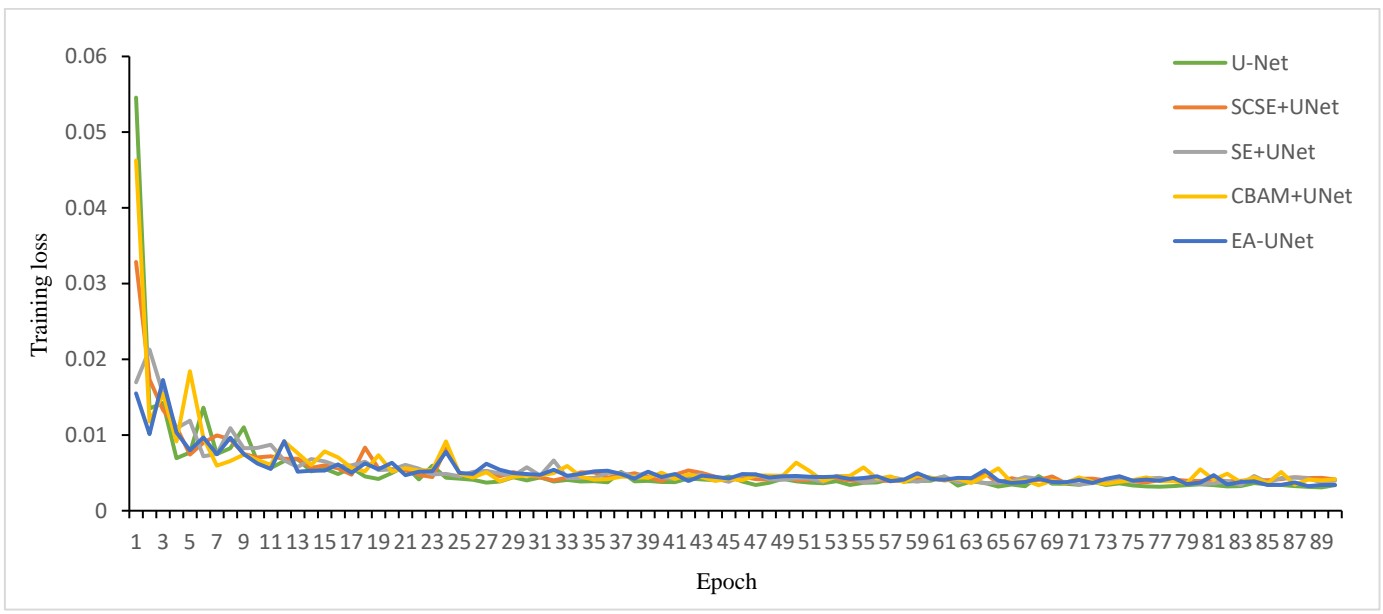

**Figure 6.** Comparison of changes in loss function for different attention modules.

### 3.3. Experiments Comparing EA-UNet Model and Other Methods

To further validate the proposed EA-UNet model's ability to segment features in the uterine cavity OCT image region, we experimentally compared it with U-Net, SegNet, DeepLabv3+, AGNet, and UNet++ (Table 3; bolded indicates the best results).

**Table 3.** Performance comparison between EA-UNet and other methods.

| Model | MIoU | Sensitivity | Specificity | Time |
|---|---|---|---|---|
| U-Net [50] | 0.8787 | 0.8886 | 0.9865 | 45.85 ms |
| SegNet [64] | 0.7998 | 0.8175 | 0.9674 | **40.05 ms** |
| DeepLabv3+ [65] | 0.8688 | 0.8823 | 0.9804 | 42.79 ms |
| AGNet [66] | 0.8862 | 0.9029 | 0.9765 | 64.14 ms |
| UNet++ [67] | 0.8727 | 0.8833 | 0.9852 | 51.71 ms |
| EA-UNet | **0.9379** | **0.9457** | **0.9908** | 50.91 ms |

The experimental results in Table 3 show that the proposed EA-UNet model has higher indexes of MIoU, Sensitivity, and Specificity compared with the comparison model. In uterine cavity OCT image segmentation, the SegNet model performs the worst, followed by DeepLabv3+ and UNet++.

To more clearly demonstrate that our proposed EA-UNet model has a more detailed segmentation effect compared with other methods, we randomly selected four images from the test set to visualize the segmentation results. Column 1 is the four original OCT images from the test set, Ground Truth indicates the true label, and columns 2 to 7 are the segmentation results of U-Net, SegNet, DeepLabv3+, AGNet, UNet++, and EA-UNet, respectively, as shown in Figure 7.

Figure 7 shows that the EA-UNet model's detail segmentation effect is better compared with all other models. The model enhances the processing ability for feature information, has a better ability to segment features, can segment regions effectively, and displays a segmentation result closest to Ground Truth. The SegNet model has the worst segmentation effect, with a serious loss of feature information. DeepLabv3+, AGNet, and U-Net models have incomplete segmentation results, display ambiguity, and ignore some boundary-detail

regions. The UNet++ model's segmentation results are second only to our proposed EA-UNet model. In summary, our analysis shows that our proposed EA-UNet model is the most effective in uterine cavity OCT image segmentation. Furthermore, we derived the loss function fitting curve shown in Figure 8 based on our experimental results. By comparing the training process of the six models, we observed that the EA-UNet model decreases the loss value faster, and the loss rate is smaller during the training process. We conclude that the improved EA-UNet model's robustness is better than the other five network models, and the experimental results also prove that the EA-UNet model's segmentation effect is better than the other five network models.

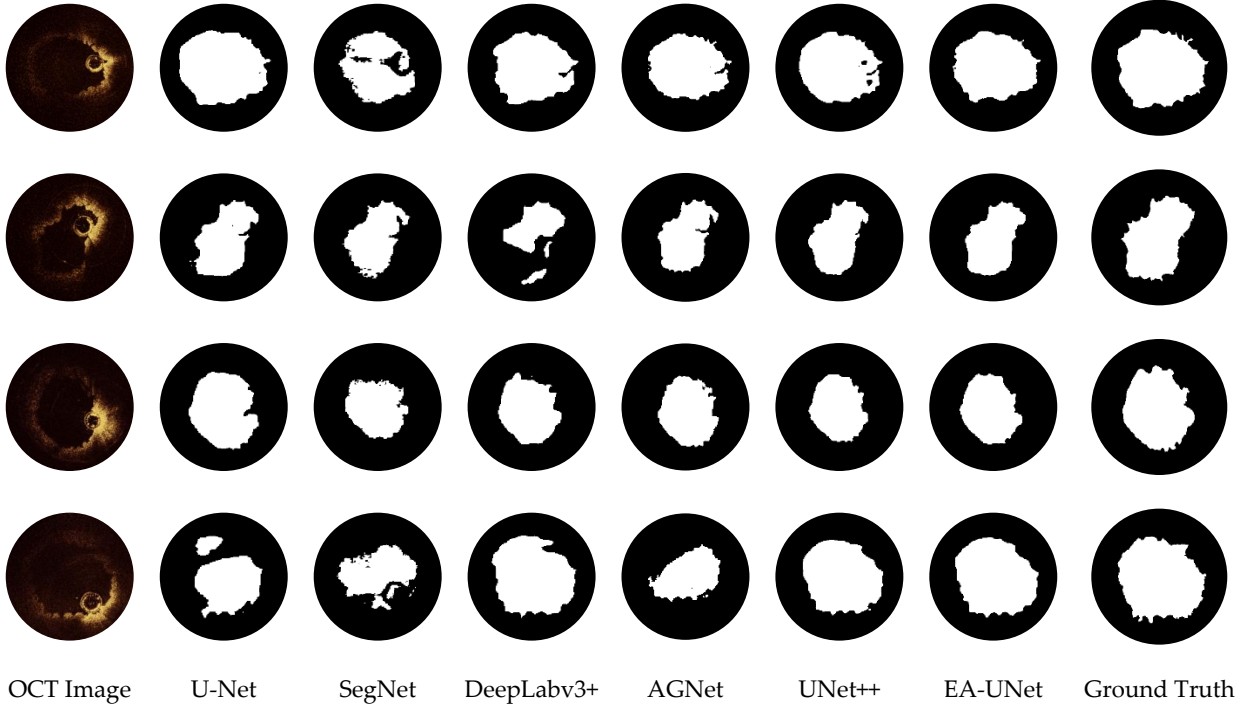

**Figure 7.** Comparison of segmentation effects of different methods.

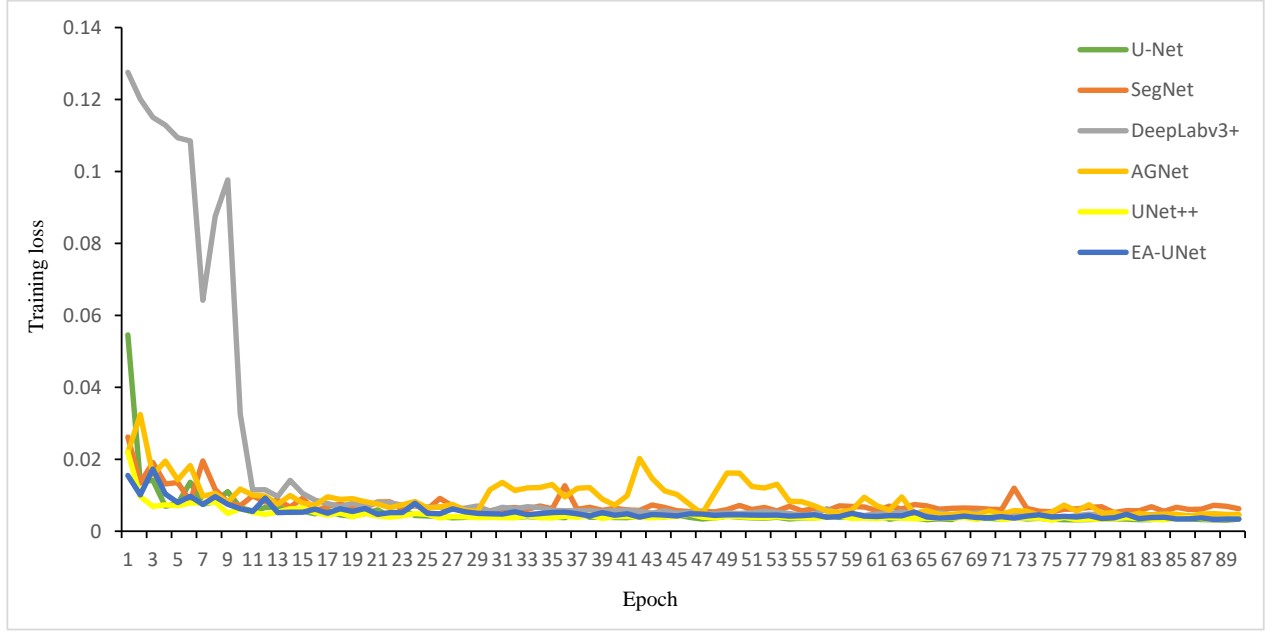

**Figure 8.** Comparison of change of loss function for different methods.

## 4. Discussion

Our proposed EA-UNet model includes the E and A modules. The E module enables the model to focus on the features in the region to be segmented and to suppress irrelevant features and extract multi-scale feature information. The A module uses global contextual information as a way to extract the detailed features of uterine cavity OCT images and improve the model's segmentation accuracy.

The aim of our study was to explore a real-time and accurate OCT image segmentation method for the uterine cavity. We used our proposed model for experiments on the OCT image segmentation of normal and damaged uterine cavities. We verified the performance of the ECA-C and Attention Gates modules in the EA-UNet model by ablation experiments. We verified the effectiveness of our proposed EA-UNet model in segmenting the uterine cavity OCT image region by comparing our experiments with other attention models. Meanwhile, we analyzed the loss function fitting curve and found that the EA-UNet model decreases the loss value faster with a smaller loss rate during the training process compared with other models. This indicates that the EA-UNet model has high robustness. Additionally, the EA-UNet model's segmentation time for a single image is 50.91ms, which also shows the model's real-time performance.

Nevertheless, the EA-UNet model's segmentation performance has much room for improvement, and we will continue to optimize the network structure to improve our model's performance. Further exploration is needed in terms of model size and loss-function construction to improve this model's accuracy for the segmentation of uterine cavity OCT images, as well as real-time performance. In OCT images, the damaged uterus's surface is discontinuous because the normal uterus's surface is continuous and smooth. Based on deep learning OCT image processing, the uterine cavity can be segmented accurately, which helps to analyze uterine cavity structure information, including thickness changes, which is important for the diagnosis of uterine cavity lesions. In future research, we will extend the proposed segmentation model to the task of grading the uterine cavity OCT images for damage at different levels. This facilitates physicians to make fast and accurate diagnoses.

## 5. Conclusions

In our study, we proposed a new EA-UNet model for the automatic segmentation of uterine cavity OCT images. First, the model uses the E (ECA-C) module, which introduces a convolutional layer on the attention mechanism ECA instead of max pool to enhance the model's feature-extraction capability, meaning the model focuses on the features in the region to be segmented and suppresses irrelevant features. In addition, we introduced the A (Attention Gates) module to utilize global contextual information as a way to extract the detailed features of the uterine cavity OCT images and improve the model's segmentation accuracy. Finally, the experimental results showed that the proposed EA-UNet model's MIoU, Sensitivity, and Specificity indexes are 0.9379, 0.9457, and 0.9908, respectively. We experimentally tested the performance of the ECA-C and AG modules and verified that the proposed EA-UNet model results in a better segmentation of uterine cavity OCT images; therefore, it can achieve the end-to-end automatic and accurate segmentation of uterine cavity OCT images.

**Author Contributions:** Conceptualization, Z.X., M.D. and J.L.; methodology, Z.X., M.D. and J.L.; software, Z.X.; validation, Z.X., M.D. and J.L.; formal analysis, M.D., X.G. and Z.C.; investigation, E.S. and J.Z.; resources, X.G.; data curation, Z.C.; writing—original draft preparation, Z.X. and J.L.; writing—review and editing, Z.X. and M.D.; visualization, Z.X.; supervision, J.Z., X.G. and Z.C.; project administration, X.G. and Z.C.; funding acquisition, X.G. and Z.C. All authors have read and agreed to the published version of the manuscript.

**Funding:** This work was supported by the National Key R&D Program of China (2019YFE0110400), National Natural Science Foundation of China (81971621, 82102087, 82102054), Key R&D Program of Hunan Province (2021SK2035), Natural Science Foundation of Hunan (2022JJ30039, 2022JJ40392),

and Clinical Research 4310 Program of the First Affiliated Hospital of The University of South China (4310-2021-K06).

**Institutional Review Board Statement:** The animal study protocol was approved by the Experimental Animal Management and Use Committee of the Shenzhen Institute of Technology, Chinese Academy of Sciences (protocol codes SIAT-IACUC-190221-YGS-ZJK-A0613 and 11 March 2021).

**Informed Consent Statement:** Not applicable.

**Data Availability Statement:** Data underlying the results presented in this paper are not publicly available at this time but may be obtained from the authors upon reasonable request.

**Conflicts of Interest:** The authors declare no conflict of interest.

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
