# Peer review of "EA-UNet Based Segmentation Method for OCT Image of Uterine Cavity"

_photonics, doi:10.3390/photonics10010073_

Round 1
Reviewer 1 Report
This paper presents a segmentation method for uterine cavity in OCT image via the improvement of the U-net. The method is sound. The experiments show the effectiveness of the proposed method. However, there are some concerns about this study.
- The motivation of this study should be further enhanced.
- The challenge of this study should be clarified.
- Please summarize and list main contributions of this study in the introduction section.
- Please clearly indicate the difference between the EA-Unet and other U-Net variants. The novelty of the proposed method should be well summarized.
- More studies on OCT image segmentation should be cited to enhance the literature review, e.g.:; Privileged modality distillation for vessel border detection in intracoronary imaging; Benchmarking automated detection of the retinal external limiting membrane in a 3D spectral domain optical coherence tomography image dataset of full thickness macular holes.
- Figure 1-3. It is better to remove background color.
- Please provide more implementation details.
- Some grammatical errors.
Reviewer 2 Report
1. In the Introduction (lines 48 to 52 of the page2), references about applications of OCT intracavitary imaging for the female reproductive tract, particularly intrauterine imaging.
2. Section 3.1 to 3.3 are not suitable for Results. Please move to Materials and Methods.
3. Were rabbit experimental procedures approved by the animal care (or regulation) committee? Please add where rabbit experimental procedures were approved.
4. In Figure 4, 5, and 7, please describe how to obtain Ground Truth images.
5. This manuscript needs Discussion. Deep learnings in this manuscript were carried out with images of normal uterine cavities. Please discuss whether this manuscript's parameters or conditions for deep learning in the abnormal (diseased) uterine cavity will be useful.
